# New Bulgarian Rootstocks for Sour Cherry Cultivars (*Prunus cerasus* L.)

**DOI:** 10.3390/plants14091352

**Published:** 2025-04-30

**Authors:** Dimitar Vasilev, Svetoslav Malchev, Lilyana Nacheva

**Affiliations:** 1Department of Plant Protection, Botany and Zoology, Konstantin Preslavsky University of Shumen, 115 Universitetska Str., 9700 Shumen, Bulgaria; 2Agricultural Academy, Fruit Growing Institute, 12 Ostromila Str., 4000 Plovdiv, Bulgaria; svetoslav.m@outlook.com (S.M.); lilyn@abv.bg (L.N.)

**Keywords:** sour cherries, rootstocks, hybrids, interspecific crossing, nursery, Argo 2

## Abstract

Research was conducted at the Agricultural Experiment Station—Khan Krum, Northeast Bulgaria during the period of 2014–2017. The aim of the study is to investigate the suitability of selected hybrids No.20-181 and No.20-192, obtained by interspecific crossing ‘Polevka’ (*Prunus cerasus* L.) × ‘Compact Van’ (*Prunus avium* L.) as clonal rootstocks for sour cherries. The rootstocks were grafted in a nursery with the cultivars ‘M-15’, ‘Nefris’, ‘Fanal’ and ‘Schattenmorelle’. *Prunus mahaleb* seedlings were used as the standard for comparison. Characteristics determining growth, the quality of the planting material and the compatibility of the rootstocks with commercial cultivars were tested. The average grafting success rate reported in the spring of the analyzed cultivar–rootstock combinations varied as follows: *P. mahaleb* (83–90%), hybrid No.20-192 (87–91%) and hybrid No.20-181 (82–85%). The selected hybrid 20-192 (‘Argo 2’) rootstock produces relatively weaker growth than the mahaleb. Hybrid 20-181 (‘Argo 1’) is characterized by the weakest growth. Both tested rootstock hybrids produce planting material with standard trunk diameter and tree height. With the weak growth that rootstock 20-181 induces in the grafted cultivar and the drought tolerance observed in 20-192, it is appropriate to continue the study in an orchard.

## 1. Introduction

Over the past 30 years, many new rootstocks, selected in different countries of the world, have entered the production of cherries. These new rootstocks serve as the basis for the introduction of new technologies and cultivars in the production of both cherries and sour cherries.

Over the past six to seven decades, breeding programs in different countries of the world have produced rootstocks of different series. This is dictated by the need for smaller trees, making them more convenient for harvesting and applying agrotechnical measures compared to traditional rootstocks from *P. mahaleb* and *P. avium*. It has not been easy to find genotypes of low-growing rootstocks within the established species, which may be the main reason for the interest in a wide range of *Prunus* species as sources for dwarf rootstocks. The incompatibility of grafts has become one of the main problems to be solved during breeding work. The most extensive breeding program of interspecific crosses turns out to be in Giessen, Germany. As results of this program, ‘GiSelA 1’, ‘GiSelA 5’, ‘GiSelA 10’, ‘GiSelA 4’ and many others were introduced into industrial fruit growing and have found their place in many countries of the world. Nowadays, sweet and sour cherry orchards are created mainly on the following rootstocks: Weiroot^®^ series, GiSelA^®^ series, ‘P-HL-A’, ‘P-HL-B’, hexaploid ‘Colt’, ‘Pi-Ku 1’, ‘Pi-Ku 4.83’, ‘Edabriz’, ‘MaxMa 14’, ‘MaxMa 60’ and ‘MaxMa 97’ [1,2,3,4,5], with recent development in innovative high-density orchards moving towards the new Corette^®^ and WeiGi^®^ series [6,7,8,9].

Choosing a suitable rootstock before establishing a cherry orchard depends on factors such as the management skills of the producer, the growth vigour of the cultivar, the planting system, and the soil and climatic conditions of the area [10]. The rootstock on which the orchard is established plays an important role in the size of the fruits, their chemical composition, the canopy volume of the tree and the resulting yield [11,12,13]. A study of cultivar–rootstock combinations in the steppe zone of the Volgograd region revealed significant variations in cultivars in terms of drought resistance and productivity, which are influenced by the rootstock used [14].

More adaptive rootstocks can be sought in the autochthonous (local) wild forms of *P. mahaleb* and *P. avium* to address climate change and ecological issues such as reduced water availability and biodiversity loss [15].

At the current stage of fruit growing science, the criteria for new rootstocks are reducing tree size, disease and pest resistance, high productivity, earlier onset of fruiting, maintaining good growth, and adaptability to changing environmental conditions [5,16,17].

Over the past two to three decades, a number of introduced dwarfing rootstocks have been tested in Bulgaria, and intensive orchards have been created using them in the country [18,19,20,21,22]. In the case of sweet and sour cherry, Bulgarian rootstocks were created in the past [23,24,25], but they do not meet modern requirements [26,27]. In this regard, the aim of the present study is to test the new selected rootstock hybrids ‘Argo 1’ and ‘Argo 2’, created at the Fruit Growing Institute—Plovdiv (part of Agricultural Academy), in a first- and second-year fruit tree nursery, in which they are grafted/budded with the most common for Bulgaria sour cherry cultivars ‘M-15’, ‘Nefris’, ‘Fanal’ and ‘Schattenmorelle’.

## 2. Results

### 2.1. Grafting (Budding) Success Rate

The success rate of grafting (or budding) and the size of the plant are crucial factors in optimizing the production of high-quality nursery plants. These traits directly affect the survival, growth rate and overall vigour of the plants, which in turn ensures increased propagation and a higher market value.

Autumn reporting, an indirect indicator, roughly estimates the expected spring survival percentage of buds (Table 1). The data show the highest percentage for the cultivar ‘M-15’ (90.8–94.6%), followed by ‘Fanal’ (88.2–91.7%). ‘Schattenmorelle’ is reported to have values of 89.3–92.1%, while ‘Nefris’ has similar results (89.2–92.7%). There are no significant differences between the four cultivars.

All cultivars budded on rootstock hybrid 20-192 (‘Argo 2’) have the highest success rate, followed by the grafts on standard rootstock Mahaleb and hybrid 20-181 (‘Argo 1’). The results depend on the rootstock, its compatibility with the cultivar, the agricultural techniques used and bud vitality.

A relatively lower percentage of grafting survival was attained during the spring reporting of buds, which is due to the lower temperatures during the winter season. The highest percentage of success rate is statistically associated with the cultivar–rootstock combination ‘Nefris’/hybrid 20-192 (91.0). It is interesting that the lowest result was also found for the cultivar ‘Nefris’ on rootstock hybrid 20-181 (82.0%).

The cultivar ‘M-15’ is distinguished by having good compatibility with all three rootstocks, and its values range from 85.0 to 90.0%. Good results were also obtained for ‘Fanal’ (83.0–88.0%). Comparatively larger differences between the autumn reporting and the spring reporting are observed in the cultivar ‘Schattenmorelle’. The obtained values for all four studied cultivars give us reason to conclude that the obtained success rate reported in the spring is good and there is no need for re-grafting (rebudding). The trend observed in the study shows that the highest percentage is consistently found in all cultivars budded on rootstock 20-192 (‘Argo 2’), followed by those on *P. mahaleb* seedlings and hybrid 20-181 (Table 1).

### 2.2. Growth Rate and Dynamics

The growth dynamics of the examined cultivar–rootstock combinations are illustrated in Figure 1. The results indicate that the four cultivars budded onto rootstock hybrid 20-181 are distinguished by having the weakest growth in terms of height. These variants show a consistent increase. From this group, ‘Schattenmorelle’/hy.20-181 exhibits the fastest growth rate among the variants. It also shows the weakest and most uniform growth of all cultivar–rootstock combinations.

The cultivars ‘Nefris’ and ‘Fanal’ on the Mahaleb rootstock, along with ‘Schattenmorelle’, ‘M-15’, ‘Nefris’ and ‘Fanal’ budded on the new hybrid rootstock 20-192 (‘Argo 2’), hold an intermediate position with respect to the growth of the one-year-old segment.

The cultivars ‘M-15’ and ‘Schattenmorelle’ on the Mahaleb rootstock have the highest trunk growth rate. The growth of the cultivated portions of the two variants proceeds in parallel, with ‘Schattenmorelle’ showing a slight predominance over ‘M-15’ until 19 June. After this date, the difference between the two increases, with ‘Schattenmorelle’ remaining predominant. This trend continues until 29 July. On the next measurement date (8 August), an alignment between ‘Schattenmorelle’ and ‘M-15’ was observed. At the 18 August reading, the cultivar ‘M-15’ showed a faster growth rate and exceeded ‘Schattenmorelle’ by about 6 cm. This trend continued up to the last report on 17 October, with a difference of 10.5 cm in favour of ‘M-15’/Mahaleb (Figure 1).

In regard to stem (trunk) diameter, ‘M-15’ budded on *P. mahaleb* seedlings is distinguished by a fast initial growth rate, which exceeds the other varieties by a significant difference up to 9 June. At the subsequent reading on 19 June (Figure 2), it aligned with and then became inferior to the ‘Schattenmorelle’/Mahaleb variant. A second equalization between the two cultivar–rootstock combinations was found during the measurement on 29 June, when a value of 10.3 mm was recorded for ‘M-15’/Mahaleb and 10.2 mm for ‘Schattenmorelle’/Mahaleb. After this checkpoint, ‘Schattenmorelle’/Mahaleb dominated over the other grafts, and on the final date, a value of 15.3 mm was recorded for it (14.7–14.9 mm).

Similar to the height indicator, the cultivars budded onto rootstock hybrid 20-192 show an intermediate trunk diameter (Figure 2). When measured early in the growing season, the four varieties showed minimal differences. On 29 June, ‘M-15’/hy.20-192 showed a higher growth rate, which continued until early August when it matched the growth rate of ‘Schattenmorelle’ and ‘Fanal’, both on hy.20-192. At the next measurement on 18 August, ‘Fanal’ and ‘Schattenmorelle’ surpassed the other two cultivar–rootstock combinations, and this dominance was maintained until the end of the vegetation period (13.1–13.3 mm).

Cultivars budded on rootstock hybrid 20-181 showed the weakest growth compared to the other two rootstocks. In this regard, at the beginning of the growing season (10 May to 9 July), there were no major differences between the four cultivars. This consistency in the dynamics of trunk diameter was maintained until the last measurement was taken.

### 2.3. Quality of Planting Material and Compliance with Standards

Table 2 presents the final values of plant height (cm) and trunk diameter (mm). The data presented indicate that the height of the trees ranges from 103.8 cm to 168.5 cm. In this regard, the combinations of ‘Fanal’, ‘Nefris’, ‘Schattenmorelle’ and ‘M-15’ with the Mahaleb rootstock are characterized as having the highest annual growth (146.2–168.5 cm). Significantly lower heights were reported for the four cultivars budded on rootstock hybrid 20-192 (134.1–145.6 cm). Plants budded on 20-181 were statistically proven to have the lowest height, ranging from 103.8 to 110.2 cm. The trees of all cultivar–rootstock combinations exceed a height of 100 cm, which meets the requirements for standard planting material according to the existing legal standards in the Republic of Bulgaria.

The stem (trunk) thickness serves as an indicator of the growth potential of the analyzed scion–rootstock combination. The trunk diameter is crucial for successful grafting in nurseries, as it ensures proper alignment and contact between the vascular tissues of the scion and rootstock, which are essential for the graft union to heal and establish a strong, healthy plant. From the data presented in the table, it is evident that the final diameter of the studied cultivar–rootstock combinations varies in the range of 11.0–15.3 mm. The highest recorded values were observed in cultivars grafted onto *P. mahaleb* seedlings, with measurements ranging from 14.7 to 15.3 mm. The cultivars budded on hybrid 20-192 (‘Argo 2’) are characterized by a thickness of 12.4 to 13.8 mm. Cultivars with the smallest trunk diameter were reported for rootstock hybrid 20-181 (‘Argo 1’) (11.0–12.1 mm). The data indicate that for all the scion–rootstock combinations, the trunk diameter exceeds 10.0 mm (Table 2), thus fulfilling the standards for quality planting material.

## 3. Discussion

The increased interest in establishing new cherry plantations necessitates the provision of new market-oriented cultivars and climate-resilient rootstocks [24,28].

### 3.1. Challenges in Rootstock Propagation and Adaptability

Ongoing rootstock research highlights a strong preference for clonal rootstocks over seedling-based propagation. This shift is driven by the complexities involved in propagating clones with specific desirable traits [5,29].

Some rootstocks with valuable traits, such as dwarfing ability and pest/disease resistance, can be difficult to propagate, necessitating advancements in propagation techniques. At the same time, adaptation to diverse environmental conditions is increasingly important, as rootstocks can extend the location adaptability of sweet and sour cherry cultivars, allowing growers to plant in suboptimal regions [24,30]. Using suitable rootstocks for propagation plays a key role in enhancing the longevity of the orchard and helps in fetching a higher market price.

While *P. mahaleb* rootstocks have been widely used for sweet and sour cherry in Bulgaria, often from seeds, there has been limited breeding work to improve their traits. Nevertheless, their adaptability to continental climates, droughts, hot summers, poor soils and lime increases their future importance, especially under climate change [29]. However, *P. mahaleb* seedling rootstocks used for cherry trees are often infected by seed-transmitted viruses and frequently show genetic incompatibility with important sour cherry cultivars such as cv. “Schattenmorelle” [31].

Clonally selected rootstocks of *P. avium* offer significant advantages for fruit growers, including uniform plant material and ease of propagation in stoolbeds for nurseries. However, these rootstocks do not enhance vigour control or precocity of scions. Conversely, interspecific hybrids with *P. avium* exhibit a wide range of levels of vigour, adaptability and disease tolerance. Among vegetatively propagated options, clones of interspecific crosses of *P. cerasus* have demonstrated the most promise [32].

### 3.2. Interspecific Hybridization and Breeding Strategies

The decline in genetic diversity among sweet and sour cherry cultivars has resulted in diminished adaptability to changing growing techniques and climatic conditions. Therefore, many breeding programs employ interspecific crosses to boost genetic diversity and introduce new traits into fruit crops [33,34]. Hybridization, specifically interspecies crossing, is frequently employed in the genetic enhancement of cherry rootstocks. Additionally, it has been utilized for the qualitative improvement of sour cherries by crossing *P. cerasus* L. (sour cherry) with *P. avium* L. (sweet cherry), although this process poses challenges due to ploidy differences [35].

According to the scientific literature reviewed by Professor Karoly Hrotko [36], cherry rootstock breeding programs, up to the year 2008, have been conducted in 17 countries, leading to the release of more than 100 rootstocks. The most extensively studied species are *P. cerasus* and the interspecific hybrid group, with 41 and 42 genotypes reported, respectively. Additionally, rootstocks originating from *P. avium* and *P. mahaleb* have each resulted in the selection of 12 genotypes [36].

The investigations into the xylem ratio within the stem cross-section [37] of ‘GiSelA 5’ revealed that this rootstock, along with *P. fruticosa* ‘Prob’, has the smallest stem porosity. Sweet cherry trees grafted onto both ‘GiSelA 5’ and ‘Prob’ rootstocks commonly face water deficiencies during hot summer days; the large difference in stem structure may influence the xylem transport and storage capacity of the trees. In contrast, *P. cerasus* ‘CAB 11E’ [37] exhibited the largest porosity. A more current investigation into ‘Oblačinska’ sour cherry genotypes revealed larger root xylem vessels, providing a high theoretical hydraulic capacity [38]. The study aimed to establish a reliable selection protocol for breeding size-controlling cherry rootstocks with potential drought adaptability, focused on the anatomical investigation of fine and skeletal roots and rootstock stems to determine their relation between growth control and drought resistance.

Recent studies of interspecific hybrids obtained from the Crimean experimental breeding station branch of the All-Russian Scientific Research Institute of Plant Growing confirm the potential of interspecific hybridization and the good compatibility of the obtained combinations, as well as the direct influence of rootstocks on the characteristics of the varieties grafted on them [14].

Over the past decade, research on cherry rootstocks has predominantly emphasized evaluation rather than the development of new rootstocks. Although there are limited new breeding initiatives and successful introductions of rootstocks, a significant number of rootstock selections are currently undergoing evaluation [36].

### 3.3. Breeding Efforts and Achievements in the Fruit Growing Institute—Plovdiv

The development of the sweet cherry breeding program, launched at the Fruit Growing Institute in Plovdiv (Agricultural Academy) in 1987, resulted in several hybrid populations from different parental combinations. The interspecific crossing of ‘Polevka’ (*P. cerasus* L.) × ‘Compact Van’ (*P. avium* L.) produced a population of 31 hybrids, out of them the two most promising (with selection numbers 20-181 and 20-192) were selected for their weak growth and field tolerance to drought and pests [39,40,41].

A study aimed at assessing the effects of different soil herbicides on the growth and health of hybrid 20-192 was conducted at the early stages of rootstock evaluation. Napropamid and Pendimethalin were found to be safe for the plants under in vitro conditions, while Metofen at high doses caused severe damage. The study highlighted the differential effects of these herbicides on plant health and growth [41]. A second study reported the content of the major mineral elements in weed species, represented in the weed association inside the nursery, as well as in the leaves of the 20-192 rootstock from herbicide-treated and untreated control variants [42].

Preliminary investigations (unpublished data) into potential drought adaptability were conducted in first- and second-year nursery at the Fruit Growing Institute—Plovdiv (FGI), Bulgaria. Hybrids No.20-181 and No.20-192 were planted in comparison to *P. mahaleb* seedlings and clonal ‘GiSelA 5’ (*P. cerasus* × *P. canescens*) to assess rootstock size, graft compatibility with the sweet cherry cultivars ‘Bigarreau Burlat’ and ‘Regina’, and survivability in non-irrigation conditions. The initial findings suggested the better survival rate and compatibility of hybrid 20-192 compared to ‘GiSelA 5’ and hybrid 20-181, while demonstrating less vigour than *P. mahaleb* seedlings. Further investigations of rootstock hybrid 20-192 (‘Argo 2’) are recommended to confirm or deny theoretical drought tolerance.

To date, the two selected hybrids, 20-181 (‘Argo 1’) and 20-192 (‘Argo 2’), have been evaluated exclusively as rootstocks for sweet cherry [41]. The current study marks the first time their potential as sour cherry rootstocks has been investigated.

## 4. Materials and Methods

The experiments were conducted in the fruit nursery of the Experimental Station of Agriculture—Khan Krum (43°12′01.6″ N 26°52′48.9″ E).

### 4.1. Origin of Rootstock Hybrids

Selected hybrids (elites) No.20-181 (‘Argo 1’) and No.20-192 (‘Argo 2’), obtained by interspecifically crossing ‘Polevka’ (*Prunus cerasus* L.) × ‘Compact Van’ (*Prunus avium* L.), were tested as clonal rootstocks for sour cherries. The clonally propagated elite genotypes were mass-produced at the biotechnology laboratory at the Fruit Growing Institute—Plovdiv, Bulgaria.

The original (source) plants reached 30-year-old trees with good vitality, grown under non-irrigation conditions and without plant protection treatments, revealing their potential for drought tolerance and resistance to diseases and pests. They were 140 cm tall (final height), with a semi-drooping canopy. They are easily propagated under in vitro conditions and with cuttings. In the initial nursery experiments, the rootstocks reached the optimal thickness at the place of grafting in the first half of September. At present, limited studies have been performed for the two hybrids in the nursery as perspective rootstocks, characterized by their weak growth vigour, suitable for both sweet and sour cherry cultivars [23].

### 4.2. Design of the Experiment

The observations were conducted in the period of 2014–2017 in first- and second-year fruit tree nursery of the research station in Khan Krum village.

The experiment was conducted using the block method, testing twelve variants (cultivar/rootstock combinations) with 100 plants as replicates—each of the four cultivars was budded onto each of the three rootstocks. The arrangement of the variants was set in long, narrow plots within a square-shaped block. The planting distances of the rootstocks were 80 × 12 cm. Standard plant protection and drip irrigation practices were applied. In mid-August, budding was carried out using the T-shaped incision method of a dormant bud. Each of the studied rootstocks was budded with the sour cherry cultivars ‘M-15’, ‘Nefris’, ‘Fanal’ and ‘Schattenmorelle’. Standard Mahaleb (*P. mahaleb* L.) seedlings were used as reference.

### 4.3. Data Collection and Analysis

The growth dynamics were recorded during the vegetation period every 10 days, with the first measurement taken on 20.05 and the last on 18.09 for each year. The trunk (stem) diameter was measured at a height of 15 cm from the soil surface using a caliper. The nursery was under irrigated conditions, and traditional agricultural techniques were applied. Data were subjected to statistical analysis using the method of multiple range test using the software product “R-4.4.2” in combination with “RStudio Desktop 2024.12.0” and installed package “agricolae 1.3-7” [43].

### 4.4. Climate Conditions During the Trial

The experimental field is located in the southern part of Shumen District at an altitude of 98 m. The main factors determining the region’s type of climate are the location of the district in the southeastern part of the Danube Plain and the possibility of unhindered invasion of northwestern, northern and northeastern air masses. The experimental area is located near the confluence of the Vrana River and the Kamchiya River. The soil in the experimental area is Luvisols [44]. The climate is temperate continental, with an average temperature sum during the growing season of 3860–3900 °C. The average amount of precipitation is 590 mm, and during the growing season, it is 370 mm. The average temperature in the period of July–August is 21–22 °C. The vegetation period is approximately 225 days, starting around 1 April and ending around 10 November. The Khan Krum microdistrict is thought to be an area extremely favourable for the development of stone fruit species.

Its climatic characteristics are influenced by many factors, such as air temperature, relative humidity, speed and direction of winds, precipitation, solar radiation, etc. The average temperature in January 2014 was 2.8 °C and the average temperature in July was 24.5 °C. The sum of precipitation in 2014 was 1078.5 mm (average for Bulgaria is 650 mm) with the summer precipitation maximum (169.5 mm) recorded in June and the autumn precipitation maximum (142.5 mm) recorded in September. The month of July is characterized by the least amount of precipitation, with 31.0 mm recorded in comparison to a norm of 60.0 mm. It should be noted that only in July was the amount of precipitation lower than normal, while the average monthly temperature was approximately 1.1 °C higher than the usual.

The recorded precipitation in the first four months of 2015 exceeded the norm. The next three months of May, June and July were characterized by relatively less precipitation than normal. The lowest average monthly air temperatures were the winter months of January and February, respectively, with 2.0 and 3.2 °C. The months of July and August were the warmest (25.8 °C), exceeding the norm by 2.4 °C. September (89.5 mm) stands out as the rainiest month of that year. The annual amount of precipitation was 615.5 mm compared to a norm of 609.9 mm, and this is the year when the recorded precipitation was the closest to the norm.

In 2016, precipitation was the most unevenly distributed. January was characterized by the highest precipitation—80.0 mm, and in February, 21.4 mm was recorded in comparison to a norm of 28.0 mm. In the first two months, the lowest temperatures were recorded, which were, respectively, 2.3 °C in January and 3.6 °C in February. It is interesting to note that during the summer months of July and August, the recorded precipitation exceeded the normal levels. The average monthly temperatures during these months are close to normal. The only month in all four years of the study during which no precipitation was recorded was September 2016. The average temperature for this month was higher than historical average by 1.2 °C. The precipitation recorded in 2016 was only 582.0 mm, making it the lowest amount out of the four years of the study.

The fourth year of the study (2017) was characterized by more precipitation than usual in January, which was equal to the norm in February. In this regard, January was the warmest relative month according to the four years of data, exceeding the norm by 1.2 °C. A decrease in air temperature was observed in February, when the average temperature was only 1.0 °C. The amount in the period from March to July was more than normal. The same trend is observed in terms of average monthly temperatures, as in the period from April to July, the air temperature was lower than normal. During the summer months of August and September, a short drought was observed, with the measured precipitation being relatively lower than normal. August and September were also characterized by average temperatures that were higher than normal. A significant amount of precipitation was also found in October (110 mm). This is the month of the year with the most recorded precipitation. In climatic terms, 660.6 mm of annual precipitation was recorded in 2017, and this amount is very close to the norm for Bulgaria (650 mm).

## 5. Conclusions

Sour cherry cultivars grafted onto rootstock hybrid 20-192 (‘Argo 2’) are characterized by a relatively high percentage of budding success in reports for both autumn and spring. The grafts on the Mahaleb rootstock exhibit an intermediate percentage, while those on the hy.20-181 rootstock demonstrate the lowest percentage. Cultivars grown on the Mahaleb rootstock tend to have the strongest growth in terms of plant height (cm) and trunk diameter (mm), followed by those on hybrid 20-192, with rootstock 20-181 showing the weakest growth. The cultivars on rootstock hybrid 20-181 exhibit a relatively weaker growth rate; however, they are evenly distributed throughout the vegetation. Simultaneously, these qualities of the hybrid could show it has potential for ornamental and urban gardening in pots. The planting material obtained from all cultivar–rootstock combinations meets the legal quality requirements adopted in the Republic of Bulgaria. Further investigations of rootstock hybrid 20-192 (‘Argo 2’) are recommended to confirm its theoretical drought tolerance.

## Figures and Tables

**Figure 1 plants-14-01352-f001:**
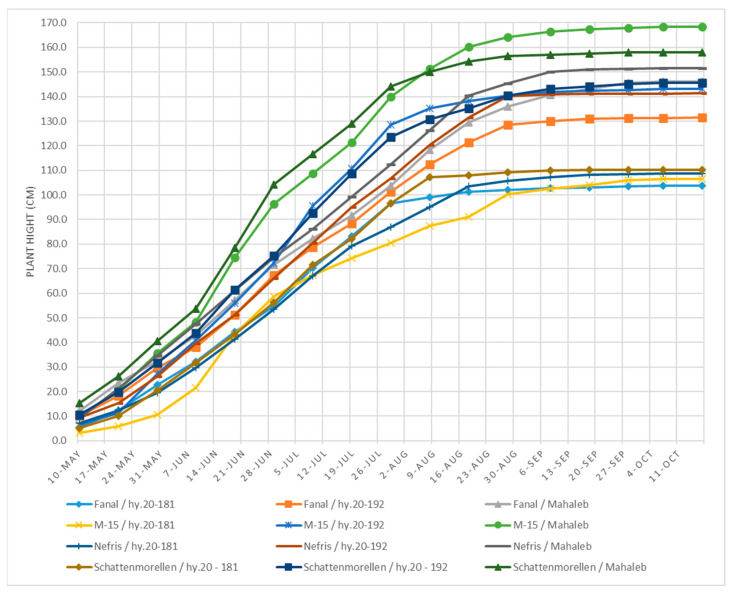
Plant height dynamics (cm)—average data for 2014–2017.

**Figure 2 plants-14-01352-f002:**
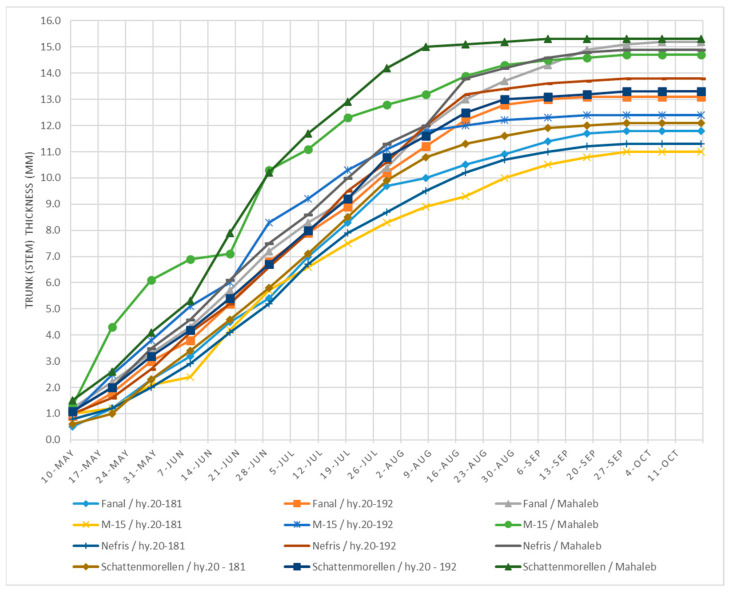
Dynamics of trunk (stem) diameter (mm)—average data for 2014–2017.

**Table 1 plants-14-01352-t001:** Budding success rate—autumn and spring reporting.

Cultivar/Rootstock	Autumn Reporting (%)	Spring Reporting (%)
Fanal/hy.20-181	88.2 d	84.1 cde
Fanal/hy.20-192	91.7 abc	88.0 abcd
Fanal/Mahaleb	90.4 bcd	83.0 de
M-15/hy.20-181	90.8 bcd	85.0 bcde
M-15/hy.20-192	94.6 a	88.9 abc
M-15/Mahaleb	93.2 ab	90.0 ab
Nefris/hy.20-181	89.2 cd	82.0 e
Nefris/hy.20-192	92.7 ab	91.0 a
Nefris/Mahaleb	90.5 bcd	88.1 abcd
Schattenmorelle/hy.20-181	89.3 cd	83.2 de
Schattenmorelle/hy.20-192	92.1 abc	87.0 abcde
Schattenmorelle/Mahaleb	90.6 bcd	86.5 abcde ^1^

^1^ Different letters in same column indicate a significant difference at *p* = 0.05.

**Table 2 plants-14-01352-t002:** Size of produced planting material—average plant size in 2014–2017.

Cultivar/Rootstock	Final Plant Height (cm)	Final Plant Diameter (mm)
Fanal/hy.20-181	103.8 d	11.8 bcd
Fanal/hy.20-192	131.4 c	13.1 abcd
Fanal/Mahaleb	146.2 bc	15.2 ab
M-15/hy.20-181	106.5 d	11.0 d
M-15/hy.20-192	143.1 bc	12.4 abcd
M-15/Mahaleb	168.5 a	14.7 abc
Nefris/hy.20-181	108.7 d	11.3 cd
Nefris/hy.20-192	141.4 bc	13.8 abcd
Nefris/Mahaleb	151.5 abc	14.9 ab
Schattenmorelle/hy.20-181	110.2 d	12.1 abcd
Schattenmorelle/hy.20-192	145.6 bc	13.3 abcd
Schattenmorelle/Mahaleb	158.0 ab	15.3 a ^1^

^1^ Different letters in same column indicate a significant difference at *p* = 0.05.

## Data Availability

The original contributions presented in this study are included in the article. Further inquiries can be directed to the corresponding author(s).

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
