# Peer review of "New Bulgarian Rootstocks for Sour Cherry Cultivars (Prunus cerasus L.)"

_plants, 2025, doi:10.3390/plants14091352_

Round 1

Reviewer 1 Report

Comments and Suggestions for Authors

The article is fascinating and useful for growers because, sometimes, incompatibility for cultivars and rootstocks can be seen till the fifth growing year. The first and second growing years in nursery conditions are sometimes quite different from orchards. 

How many seedlings did you use from each variant? Did you see growing results after the nursery?

Author Response

The article is fascinating and useful for growers because, incompatibility for cultivars and rootstocks can be seen till the fifth growing year. The first and second growing years in nursery conditions are sometimes quite different from orchards. 

Coments 1. How many seedlings did you use from each variant ? Did you see growing results after the nursery ?

Point 1. For every variant (combinations of rootstock and cultivar) in the study, 100 plants have been planted and budded.

Coments 2. Did you see growing results after the nursery ?

Point 2. No, the study and the results were focused on the production of standard planting material. Future studies are planned (an orchard is created with some of the produced trees) to evaluate the rootstocks performance in production orchard. 

Reviewer 2 Report

Comments and Suggestions for Authors

The manuscript evaluates new Bulgarian rootstock hybrids for sour cherry cultivars, focusing on their grafting success rates and growth performance. The results show promising potential for use in cherry orchards, particularly for drought tolerance. The manuscript required extensive revision before publications.

  1. Results:

Structure the results section and report the results separately according to different analysis indicators

2.The experimental data on drought tolerance in the results were not presented. How to draw the conclusion of drought tolerance?

3.Title:

The title does not clearly state the relationship between New Bulgarian Rootstocks and Sour Cherry Cultivars. It is recommended to include an evaluation or analysis in the title, such as "Evaluation of New Bulgarian Rootstock Hybrids for Sour Cherry Cultivars (Prunus cerasus L.)"

4.Materials and Methods:

Provide a detailed description of the experimental design, sample size, number of repetitions, indicator measurement, data collection, and analysis methods to improve the reproducibility of the study.

5.line 288, there appears 2.1 Climate conditions during the trial

6.Introduction:

Introduction should be more constructive with rationale of the study. Elaborate clearly, why this research is necessary

  1. Discussion:

The section lacks organization and logical relationships. If necessary, subheadings can be added. 8.Truthfully discuss the limitations and shortcomings of the research, propose future research directions and improvement suggestions.

9.The conclusion section should summarize the main findings and significance of the study in a concise and clear manner, avoiding duplicating the content of the results section.

Author Response

Point 1: Structure the results section and report the results separately according to different analysis indicators.

Response 1: Yes, we agree with the reviewer"s funding. As recommended by both Reviewer 2 and Reviewer 3, Subtitles were introduced in section "Results'' to improve structure and  clarity.

Point 2: The experimental data on drought tolerance in the results were not presented. How to draw the conclusion of drought tolerance ?

Response 2: In section "Discussion" we have mentioned preliminary unpublished data in order to present trends that were observed in the nursery. There trends need futher investigation in an orchard and more precise measurements. Such study is planned, and orchard is created. We have clearly indicated that these are preliminary unpublished data. 

Point 3: Title: The title does not clearly state the relationship between New Bulgarian Rootstocks and Sour Cherry Cultivars. It is recommended to include an evaluation or analisys in the title, such as "Evaluation of New Bulgarian Rootstock Hybrids for Sour Cherry Cultivars (Prunus cerasus L.)''

Response 3: The title is simple enough stating that these are new rootstocks for sour cherries created in Bulgaria. Adding "Evaluation of'...'' will shift the focus from ''...To date, the two selected hybrids, 20-181 (''Argo 1'') and 20-192 (''Argo 2''), have been evaluated exclusively as rootstocks for sweet cherry [41]. The current study marks the first time their potential as sour cherry rootstocks has been investigated.....''

Point 4: Materials and Methods: Provide a detailed description of the experimental design, sample size, number of repetitions, indicator measurement, data collection, and analysis methods to improve the reproducibility of the study.

Response 4: The authors agree with the reviewer's commend and more details have been added in section "Materials and Methods''

Point 5. line 288, there appears 2.1 Climate conditions during the trial

Response 5: Yes, even though that drip irrigation was applied in the nursery, temperature and rainfall still play an important role not only the growth of the rootstocks and later the scions, but also for the success rate of the budding.

Point 6:  Introduction should be more constructive with rationale of the study. Elaborate clearly, why this research is necessary. 

Response 6:  The authors believe that in several places in the manuscript the rationale of the study was stated. An example: '' ...current study marks the first time their potential as sour cherry rootstocks has been investigated...'' In the case of sweet and sour cherry, there are Bulgarian rootstocks created in the past, but they do not meet the modern requirements...'', ''...increased interest in establishing new cherry plantations necessitates the provision of new market-oriented cultivars and climate resilient rootstocks...''

Point 7:   Disscusion: The section lacks organization and logical relationships. If necessary, subheadings can be added ?

Response 7: The authors agree with the reviewer's commend and corrections have been applied to the manuscript. Subtitles were introduced in section ''Discussion''to improve structure and clarity. 

Point 8:  Truthfully discuss the limitations and shortcomings of the research, propose future research directions and improvement suggestions.

Response 8: The present research is limited to nursery and production of planting material (trees). In section ''Discussion" we have mentioned preliminary unpublished data in order to present trends that were observed in the nursery. There trends need futher investigation, but in an orchard and more precise measurements. Such study is planned and orchard is created. 

Point 9: The conclusion section should summarize the main findings and significance of the study in a concise and clear manner, avoiding duplicating the content of the results section.

Response 9: The authors agree with the reviewer's commend and corrections have been applied to the manuscript.  

Reviewer 3 Report

Comments and Suggestions for Authors

This research on new rootstock potential for sour cherry cultivars presents valuable findings on plant growth measurements. While the manuscript is well-organized overall, I suggest the following improvements:

  1. For Figures 1 and 2, please clearly indicate what the y-axis represents (e.g., height in cm, diameter in mm).
  2. Scientific names should follow proper convention: write the full name at first mention (e.g., Prunus cerasus) and use abbreviated form thereafter (e.g., P. cerasus).
  3. Consider incorporating additional physiological measurements such as chlorophyll content or stomatal cross-sections to strengthen your findings.
  4. Expand the discussion section to address why different plant cultivars exhibited variations in height and diameter. Include an analysis of their genetic backgrounds and physiological differences that might explain these growth variations.
  5. Restructure the manuscript by moving content from section 2.1 (Materials and Methods) to the Results section and summarize this information in a table for clarity.
  6. Consolidate the Conclusion section into a single, comprehensive paragraph that synthesizes all key findings.

Author Response

The research on new rootstock potential for sour cherry cultivars presented valuable findings on plant growth measurements. While the manuscript is well-organized averall, I suggest the following improvements:

Point 1: For Figures 1 and 2, please clearly indicate what the y-axis represents (e.g., height cm, diameter in mm).

Response 1: The authors agree with the reviewer's commend and corrections have been applied to the manuscript. 

Point 2: Scientific names should follow proper convention: write the full name at first mention (e.g., Prunus cerasus) and use abbreviated form thereafter (e.g., P. cerasus)

Response 2: The authors agree with the reviewer's commend and corrections have been applied to the manuscript. 

Point 3: Consider incorporating additional physiological measurements such a chlorophyll content or stomatal cross-sections to strengthen your findings.

Response 3: The focus of the study is to test whether the rootstocks have good graft compatibility and if they are suitable for production of standard planting (trees) from sour cherry (Prunus cerasus L.). Nevertheless, the reviewer's suggestion is taken into consideration for the fututre studies that have been planted in a production orchard after the nursery. 

Point 4:  Expand the discussion section to address why different plant cultivars exhibited variations in height and diameter. Include an analisys of their genetic backgrounds and physiological differences that might explain these growth variations.

Response 4: The authors understand and agree with the reviewer's point of view, but the genetic background of the new rootstock is presented in details in section ''Materials and Methods''. Adding in section ''Discussion"explanation for the vigorous Mahaleb rootstock, used as reference (standard) in numerous studies in the past, will only shift the focus away from the new hybrids and will not contribute with new knowledge or insight 

Point 5: Restructure the manuscript by moving content from section 2.1 (Materials and Methods) to the Results section and summarize this information in a table for clarity.

Response 5: The authors believe that these are experimental (field) conditions that are determined by the trail location and region's climate type, and they are more suited in the section ''Materials and Methods". And even though that drip irrigation was applied in the nursery, temperature and rainfall still play an important role not only the growth of the rootstocks and later the scions, but also for the success rate of the budding.

Point 6:  Consolidate the Conclusion section into a single, comprehensive paragraph that synthesizes all key findings.

Response 6: The authors agree with the reviewer's commend and corrections have been applied to the manuscript. 

Round 2

Reviewer 2 Report

Comments and Suggestions for Authors

The author adjusted the structure of the manuscript to make it clearer. At the same time, the author made revisions and improvements to the details in the manuscript. Therefore, I believe that the manuscript version can be published.

Reviewer 3 Report

Comments and Suggestions for Authors

The authors have responded to almost all of the comments or suggestions.